# Structured Routine Use of Styletubation for Oro-Tracheal Intubation in Obese Patients Undergoing Bariatric Surgeries—A Case Series Report

**DOI:** 10.3390/healthcare12141404

**Published:** 2024-07-15

**Authors:** Hsiang-Chen Lee, Bor-Gang Wu, Bo-Cheng Chen, Hsiang-Ning Luk, Jason Zhensheng Qu

**Affiliations:** 1Department of Anesthesia, Hualien Tzuchi Hospital, Hualien 97002, Taiwan; 102311117@gms.tcu.edu.tw; 2Department of Surgery, Hualien Tzu-Chi Hospital, Buddhist Tzu Chi Medical Foundation and School of Medicine, Tzu-Chi University, Hualien 97002, Taiwan; brogen@tzuchi.com.tw; 3Department of Otolaryngology, Hualien Tzu Chi Hospital, No. 707, Sec. 3, Chung-Yang Road, Hualien 97002, Taiwan; berryabc127@tzuchi.com.tw; 4Laboratory of Bio-Math, Department of Financial Engineering, Providence University, 200, Sec. 7, Taiwan Boulevard, Shalu Dist., Taichung City 43301, Taiwan; 5Department of Anesthesia, Critical Care and Pain Medicine, Massachusetts General Hospital, Harvard Medical School, Boston, MA 02115, USA; jqu@mgh.harvard.edu

**Keywords:** styletubation, video intubating stylet, tracheal intubation, oro-tracheal intubation, laryngoscope, laryngoscopy, videolaryngoscope, anesthesia, difficult airway, airway management, paradigm shift, new paradigm, obesity, bariatric surgery

## Abstract

The aim of this case series report is to provide a new topical view of styletubation (video intubating stylet technique) in obese patients undergoing bariatric surgeries. In contrast to various conventional direct laryngoscopes (DLs), videolaryngoscopes (VLs) have been applied in such obese populations with potentially difficult airway complications. The safety and effectiveness of VLs have been repeatedly studied, and the superiority of VLs has then been observed in and advocated for routine use. In this article, among our vast use experiences with styletubation (more than 54,998 patients since 2016) for first-line routine tracheal intubation, we present the unique experience to apply the styletubation technique in obese patients undergoing bariatric surgery. Consistent with the experiences applied in other patient populations, we found the styletubation technique itself to be swift (the time to intubate from 5 s to 24 s), smooth (first-attempt success rate: 100%), safe (no airway complications), and easy (high subjective satisfaction). The learning curve is steep, but competency can be enhanced if technical pitfalls can be avoided. We, therefore, propose that the styletubation technique can be feasibly and routinely applied as a first-line airway modality in obese patients undergoing bariatric surgery.

## 1. Introduction

Obesity has been a worldwide medical disease and bariatric surgery is the most effective therapy and sometimes the last effort to lose body weight. Modern bariatric procedures, together with other usual medical and lifestyle treatments, have demonstrated effectiveness and safety [1]. Anesthesia and peri-operative management for bariatric surgeries in all categories of obese patient populations have therefore become the issues for discussion [2,3,4,5]. Among all the related peri-operative care issues for bariatric surgery, airway management in such obese populations has been extensively discussed [6,7]. Among all the modalities of airway management, oro-tracheal intubation remains the most challenging part in obese patients undergoing bariatric surgery [8,9].

The excessive body fat deposition externally and particularly around the upper airway (neck and oro-pharynx) in obese patients tends to make oro-tracheal intubation worse. Proper pre-operative planning and preparation of airway management are crucial, including ramp positioning, adequate pre-oxygenation, and depth of anesthesia and analgesia [10,11,12]. Whether awake intubation, rapid sequence induction and intubation (RSII), or routine oro-tracheal intubation be the choice of airway modality depends on many factors clinically, including patient’s factors, airway operator’s experiences and competencies, environmental issues, and facility support (for review, see [8,13,14]). The prevalence rate of difficult tracheal intubation (DTI) in obese patients is 3.3% defined by three or more attempts and 8.3% defined by Cormack–Lehane views 3 and 4 [15]. Such, the prevalence rate of DTI in obese patients, if defined differently (e.g., based on the intubation difficulty scale (IDS) score), can be different, e.g., 15.5% [16] and 13.8% [17].

While difficult, tracheal intubation is more common among obese patients [18,19]; none of the classic risk factors for DTI were satisfactory to be predictive in obese patients [16,20,21,22,23,24]. Conflicting results did not support any definitive roles of certain predictors for problematic or difficult intubation, e.g., BMI, neck circumference (NC), modified Mallampati score (MMS), mouth-opening, thyromental/sternomental distance (TMD/SMD ratio), range of motion of neck, etc. For obese population presented with certain difficult airway (DA) predictors, a videolaryngoscope (VL) is believed to be applicable for tracheal intubation and superior to the conventional direct laryngoscope (DL). However, among all the common outcome parameters (i.e., better overall satisfaction score, intubation time, number of intubation attempts, and necessity of use of extra adjuncts), the VL shows longer intubation time in addition to better laryngopscopic views and lower IDS scores [25,26,27,28,29,30].

It is important to note, however, that objective evidence from large randomized clinical trials to support the use of VLs as a first-line airway management modality in obese patients is still lacking [31]. In contrast to the routine use of VLs, we have universally applied styletubation (i.e., video intubating stylet technique) as a routine first-line oro-tracheal intubation modality in our medical center (8049 cases out of 8329 tracheal intubating cases in 2023, Tzu Chi hospital, Taiwan). In this case series article, we present our experience of applying styletubation in 20 obese patients undergoing bariatric surgeries. In particular, the styletubation technique is feasible even when such obese patient populations exhibited certain formidable clinical scenarios which might complicate airway management.

## 2. Case Presentation

We are equipped with several different commercial products of video intubating stylets for tracheal intubation in our operating rooms. The devices include C-MAC^®^ VS (Karl Storz GmbH & Co. KG, Tuttlingen, Germany); UE video stylet (UE, Xianju, Zhejiang, China); TuoRen Video Intubating System (Henan Tuoren Medical Device Co., Ltd., Xinxiang, China); Stylet-Go Mini Plus Video Stylet scope (Anestek Corp, Taoyuan, Taiwan); and Trachway video intubation system (Markstein Sichtec Medical Corp., Taichung, Taiwan). The styletubation technique is conducted with a video intubating stylet [32,33,34,35]. In brief, the video intubating stylet device is a rigid (or semi-rigid) stylet equipped with a CMOS camera at its tip projecting to a separate or attached monitor. The endotracheal tube (ETT) is then mounted onto the video stylet in a similar manner to a traditional stylet (Figure 1A, upper panel). The tip of the stylet is designed to be malleable and/or anteflex. Because the camera resides in the stylet, the proper-positioned video stylet can then acquire a clear image in front of the stylet–ETT unit (Figure 1A, lower panel).

There are several manners to perform styletubation for tracheal intubation. The most common and easy example is the two-person model, shown in Figure 1B,C. The airway management assistant either stands alongside or face-to-face with the airway operator, and is dedicated to open the patient’s mouth and to perform a jaw-thrust maneuver. Otherwise, when there is only one operator, styletubation can also be performed with the Shikani technique [36] or facilitated with a laryngoscope [37]. The original Bonfils-type intubating stylet with a retromolar approach technique is to serve as an alternative rescue airway tool in patients with a very limited mouth opening. In the present presentation, we always adopted a midline approach which served very well in obese patients without a scenario of trismus or oro-pharyngeal cancers.

Patients shown in Table 1 were managed by the same anesthesiologist (HN Luk) with an expertise in anesthesia for bariatric surgery. The following standard anesthesia technique was applied: Routine high-flow pre-oxygenation before induction was performed. Either a high-flow nasal cannulation (30 L/min) or a face mask ventilation (15 L/min) was applied for pre-oxygenation. Patients were positioned in a stacking ramp with a reverse Trendelenburg positioning of 25 degrees. Standard vital signs monitoring was implemented, including non-invasive blood pressure monitoring, arterial line, pulse oximeter (SpO2), bispectral index—BIS or density spectral array—DSA, capnography (ETCO2), neuromuscular monitoring (train-of-four and post-tetanic contractions), pain intensity monitoring (surgical pleth index—SPI or analgesia nociception index—ANI), hemodynamic monitoring (FloTrac), and body temperature. Glycopyrrolate and lidocaine were first administered. Then, anesthesia was induced with midazolam (2.5 mg), ketamine (25 mg), fentanyl (100 μg), propofol (using the adjusted dose formula: adjusted dose = ideal body weight (IBW) dose × [1 + 0.007(total body weight − IBW)]), and rocuronium. A two-hand mask ventilation technique by an airway assistant was performed. Before oral tracheal intubation was conducted, oral suction with a modified airway-suction tube unit was used to clear any possible saliva or secretions along the oro-pharyngeal space. The patient’s mouth was then opened by an airway assistant with an effective jaw-thrust maneuver to lift up the patient’s epiglottis. Then, styletubation was performed with a video intubating stylet by the airway operator (Figure 1B). Anesthesia was then maintained with both inhalational anesthetic (sevoflurane or desflurane, around 0.5 MAC) and propofol (ICI, effect concentration around 1.5 μg/mL). A neuromuscular blockade was maintained with rocuronium and monitored with ToF and PTC indices. During emergence, rocuronium was chelated by sugammadex. Tracheal extubation was performed when ToF reached > 0.95 and patient could obey oral command. Post-operative CPAP was used if desaturation or CO_2_ retention occurred. When necessary, the patient was kept in the ICU overnight.

Among 20 patients listed in Table 1, we present several clinical scenarios when styletubation is applied in obese patients undergoing bariatric surgeries. In particular, the styletubation technique is feasible even when such obese patient populations exhibited certain outstanding clinical conditions which might complicate airway management. Patients undergoing bariatric surgery may have varied degrees of obesity. Figure 2 shows such an example of a patient with a BMI of 34.8 kg/m^2^. The styletubation procedure was easy and smooth in this category of patients. The intubating time (from lip to trachea) was 5 s with first-pass success. In contrast, in patients with much higher BMIs (e.g., above 60 kg/m^2^), pre-operative airway evaluation may predict potential DTI (e.g., high degrees of Mallampati score, bull-necked, increased neck circumference, accumulation of adipose tissue around upper airway, limited neck extension, etc.). Figure 3 shows such an example of super-super obesity. During styletubation for tracheal intubation, more pronounced collapsed pharyngeal structures, omega-shaped epiglottis, and limited visualization of glottis were seen in this patient (BMI > 100 kg/m^2^). Even so, the oro-tracheal intubation by styletubation was smooth, easy, prompt (intubation time: 12 s), with first-pass success.

The obese patients commonly presented themselves with either a pear-shaped or apple-shaped body habitus which was reflected by the over-accumulation of fat in the abdomens, waists, hips, and thighs. With higher chance, the fat pad also caused narrowing or collapse of the pre-tracheal and pharyngeal space. While in the scenario of the obese patient under anaesthetized and neuromuscular blockade conditions, the pharyngeal folds/walls might be weighed down by excess fat around the upper airway and therefore prevent clear visualization of the larynx. Figure 4 shows such an extraordinary scenario in obese patients undergoing bariatric surgery. In addition to an omega-shaped folded epiglottis (Figure 3E,F), the epiglottis might also be floppy and collapsed (Figure 4A,B). Also, exaggerated hypertrophy and edema over arytenoids and vestibular folds might severely obstruct the view of true vocal cords in obese subjects (Figure 4C,D).

One of the predicted DTIs is in a patient with limited cervical spine mobility, which can cause more difficulties in obese patients. Here we present a 56-year-old man with a past history of cervical spondylosis with radiculopmyelopathy and quadriparesis. A surgical history of anterior cervical discectomy and fusion (ACDF) in C3/C4, C4/C5, and C5/C6, with PEEK cage fusion and Zephir plate fixation was noted. Other medical history included hypertension, type 2 diabetes mellitus, old cardiovascular accident, and obstructive sleep apnea. His BMI was 45.4 kg/m^2^ and underwent LSG and mesh repair of the ventral hernia. Pre-operative airway evaluation showed a higher degree of Mallampati grading, a short neck, and less cervical extension (Figure 5A–D). During the styletubation procedure, copious saliva and secretions, collapsed airway soft tissues, and a dropped and folded epiglottis were observed (Figure 5E,F). Even in the presence of unfavorable conditions for oro-tracheal intubation in such a case scenario, the performance of styletubation was still smooth (first-pass success) and swift (intubating time: 13 s).

It is important to know whether a less-experienced airway manager performing styletubation is still competent under potential DTI scenarios like those in obese patients undergoing bariatric surgery. Figure 6 shows such a performance of styletubation by a third-year anesthesia resident in an obese patient undergoing bariatric surgery. The intubation was swift and smooth with first-pass success.

One of the inherent problems of optical devices for tracheal intubation is the impact of patient secretions and saliva on the acquired visual field. Figure 7 shows such impact of secretions on styletubation in two obese patients undergoing bariatric surgery (Figure 7A–F). Foaming occurs when excessive saliva/secretions is pooled in the mouth and throat. Such bubbles might obstruct view during intubation. In obese patients undergoing bariatric surgeries, an adequate and effective suctioning method is therefore in particular helpful and crucial. Figure 8 shows such an example using a modified suction tube–airway tube unit to clear the airway before styletubation commenced.

In contrast to the application of DL or VL for tracheal intubation in obese patients, styletubation could be facilitated by laryngoscopy in the obese patient population. Figure 9 shows such a combined styletubation with VL in an obese patient undergoing bariatric surgery. Both the first-pass success and time to intubation are excellent with this combined technique, especially useful for a novice airway operator to learn. The ancillary role of laryngoscopy in styletubation is in particular useful and helpful in intubating obese patients whose upper airway soft tissues might be exaggeratedly collapsed and congested.

## 3. Discussion

In the present presentation, we demonstrate the effectiveness and safety of the styletubation technique for tracheal intubation in 20 obese patients undergoing bariatric surgeries. As shown in Table 1, the mean BMI is 38 kg/m^2^ (ranged from 103 to 34.8, with a median of 48) which is comparable to those in the literature. With such a styletubation technique, the mean intubating time (from lip to trachea) is 12 s (ranged from 5 to 22, with a median of 11) which is much shorter than those reported in the literature. Notably, the first-pass success rate is 100% with a perfect POGO (percentage of glottic opening) scale, and no airway-related adverse events or complications occurred during the styletubation procedure.

It is known that more than half of all airway incidents and severe/serious complications follow primary difficulties during tracheal intubation. Such sub-optimal attempts at laryngoscopy and intubation usually are continuous wasted attempts that might eventually lead to clinical catastrophic disasters [38]. In this respect, VL shows a much better chance and possible superiority over DL under various difficult airway scenarios and in critically ill patients [25,39,40]. Therefore, VL has been called for use as a first-line airway tool for routine tracheal intubation and to be accessible at all times, including in application in obese patients [41,42,43,44]. In contrast to the potentially dominant and superior role of VL in the field of airway management, the emerging role of styletubation for tracheal intubation might be in a position to challenge the already leading position of VL in the near future [33,34,35].

### Limitations

Several limitations and technical pitfalls of VL have been presented and discussed [45]. In addition to the perils of use of the styletubation technique, the pitfalls of the intubating stylet (e.g., Bonfils-type stylet) have been described earlier [46]. The commonly identified disadvantages of such a rigid optical airway management device are as follows: (1) losing spatial orientation or failing to acquire a clear and identified visualization of oro-pharyngeal structures; (2) anatomically structural images becoming blurred and un-identifiable because of the close contact of the scope lens abut to the airway anatomical structures or secretions/blood (e.g., in Figure 5 and Figure 7); (3) inadequate lift-up of the epiglottis by manual jaw-thrust alone and therefore causing sub-optimal glottis visualization. Other human engineering factors include (1) the poor coordination of the intubator’s eye/wrist/arm movements; (2) the absence of a habit to identify the signposts/landmarks along the airway pathway (e.g., palate, uvula, tongue base, epiglottis, vocal cords); (3) the lack of proper prevention or minimizing fogging over the lens; (4) the lack of a functional stopper device on to the proximal end of the stylet to hold the ETT in a proper position.

As those “did not work” problems of the endoscopy technique, the suboptimal imaging quality and invisibility of regions of interest (e.g., epiglottis, vocal cords, anterior commissure, etc.) similarly occurred when the styletubation technique was applied in obese patients. Possible troublesome scenarios include the following: oro-pharyngeal accumulation of saliva/secretions, retention and pooling of copious secretions nearby glottis, diffusely blurred images due to fogging of the lens, and partially blurred image due to secretions onto the lens itself (e.g., in Figure 5 and Figure 7). Such scenarios could prolong intubation time or cause failure to intubate. A practical solution is to routinely use an improvised suction apparatus (i.e., an airway-suction tube unit) and effectively suck all secretions out beforehand, also shown in Figure 2, Figure 3 and Figure 4 [47]. Such a suction unit not only can help clearing the secretions, but also can serve as a visual guide to lead the stylet scope into the epiglottic region more easily (Figure 8).

The superior roles of VL in patients with limited cervical spine mobility have been widely documented [48,49,50,51]. The advantages of VL include causing less upper cervical spine motion, better probability of first-pass success, and less risks of intubation failure during cervical immobilization or with limited cervical spine mobility. Recently, the application of styletubation has also been demonstrated in such potential DA scenarios [52,53]. The advantages of styletubation include swift intubating time, excellent first-pass success rate, and less airway stimulation or injuries. Similarly, in obese patients, complicated by limited cervical spine mobility, undergoing bariatric surgery, DA would be expected. The advantageous role of styletubation is shown in Figure 5.

In comparison to the learning curve of using DL by novice trainees, VL shows itself as safer and more usable for the unexperienced airway providers and learners, especially for difficult airway scenarios [54,55,56]. Again, styletubation also provides such advantages for the learners (Figure 6) and is easy to learn [57,58]. Although a better laryngeal view was obtained than with DL, the inherent problem with VL is the difficulty in landing the endotracheal tube into the trachea correctly and smoothly (possibly attributes to thickness and width of the blades, and the more acute angle of the VL’s view). The Bonfils-type endoscope, however, was also found to be more difficult to intubate in an obese patient, mostly due to redundant or collapsed soft tissues impeding the glottic view. Fortunately, to circumvent extraordinary airway collapsibility in obese patients, the ancillary role of VL (or DL) to assist styletubation in one such DA scenario has been demonstrated in Figure 9. Similar advantages of combing styletubation with DL/VL have been previously reported [59,60,61].

There are inherited limitations to the present case series report. In contrast to the highest level of evidence of the prospective randomized clinical trials, however, we believe this case series report may still provide a new insight into the unique airway management modality in obese patients undergoing bariatric surgery, and serve as a source of quick up-to-date reference for readers with an interest in styletubation and laryngoscopy. Although we have a vast array of clinical case experiences of applying the styletubation technique (more than 50,000 cases since 2016), we have not yet conducted a prospective comparative study of styletubation against laryngoscopy in obese patients undergoing bariatric surgery. Some of the practical difficulties of such a robust study design include different competencies and preferences of the airway operators, uncontrolled clinical conditions and scenarios, ethical issues involving human subjects, etc. While we present 20 clinical cases for using styletubation in this article, the sample size is small, and it is impossible to make any statistical analysis on its significance. Also, this clinical cases report was conducted mainly by a very small group of airway operators in a single medical center. Therefore, the generalizability of the study conclusion may be limited, and the use and interpretation of the results should be carefully considered to avoid any unnecessary overstatements.

## 4. Conclusions

The success of tracheal intubation by a variety of airway intubating tools is dependent on many factors, including the training programs, skills, and competency of the airway operator. Meanwhile, the basic tenets of tracheal intubation remain to be “first-attempt success” and “intubation time”, both even more crucial in obese patients undergoing bariatric surgery. We have demonstrated, in this article, that the styletubation technique serves well for one such goal of tracheal intubation in this particular clinical scenario (Table 1). With the increasing attention on the styletubation technique in various clinical scenarios [32,33,34,35,62,63,64,65], we believe the potential clinical role of styletubation will eventually supersede that of laryngoscopy in the near future. In conclusion, in obese patients undergoing bariatric surgery, the role of the routine application of styletubation is promising and worthy to test.

## Figures and Tables

**Figure 1 healthcare-12-01404-f001:**
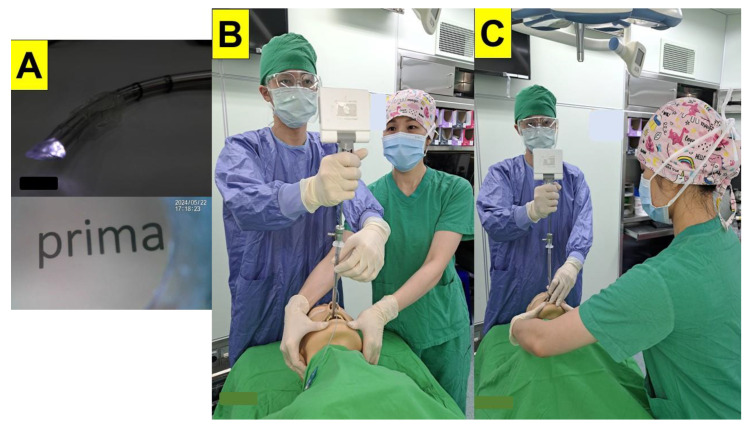
The styletubation (video intubating stylet technique). (**A**) The video intubating stylet–endotracheal tube unit (upper panel) and an inside-out clear view acquired from the camera at the tip of the stylet (lower panel). An example of the two-person model to operate the styletubation. The airway assistant either stands should-to-shoulder with the operator (**B**) or face-to-face with the operator (**C**). The assistant ensures effective mouth-opening and jaw-thrust on the obese patient.

**Figure 2 healthcare-12-01404-f002:**
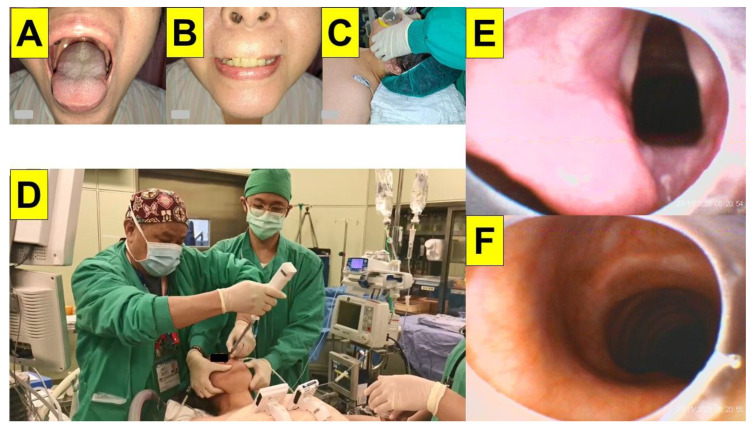
An example of an obese patient with BMI 34.8 kg/m^2^ (Case-20, Table 1) undergoing bariatric surgery. Pre-operative airway evaluation included mouth-opening (**A**) and upper-lip bite test (**B**). (**C**) Mask ventilation with jaw-thrust. (**D**) Clear visualization of glottis (**E**) and trachea (**F**). Time to intubate: 5 s, and succeeded in the first attempt. (See Appendix A).

**Figure 3 healthcare-12-01404-f003:**
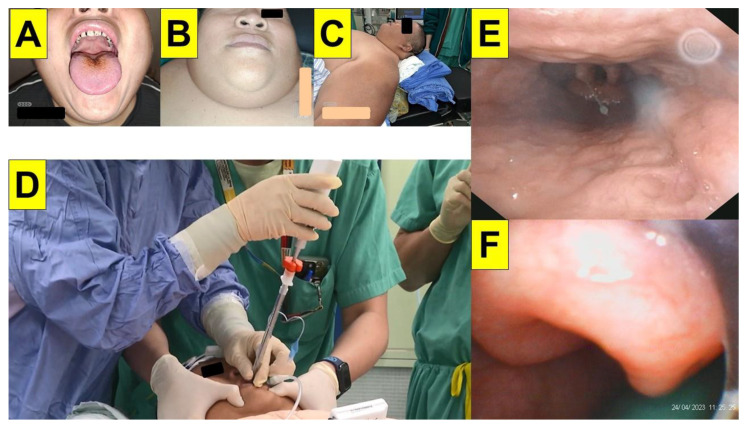
An example of a super-super obese patient with BMI 103.0 kg/m^2^ (Case 1, Table 1) undergoing bariatric surgery. (**A**) Pre-operative airway evaluation included mouth-opening. (**B**) Excessive fat tissue deposition around the neck. (**C**) Reverse Trendelenburg position. (**D**) Styletubation with two-person model. (**E**) Pharyngo-laryngeal view. (**F**) Close-up view of the omega-shaped epiglottis. Time to intubate: 12 s, and succeeded in the first attempt. (See Appendix A).

**Figure 4 healthcare-12-01404-f004:**
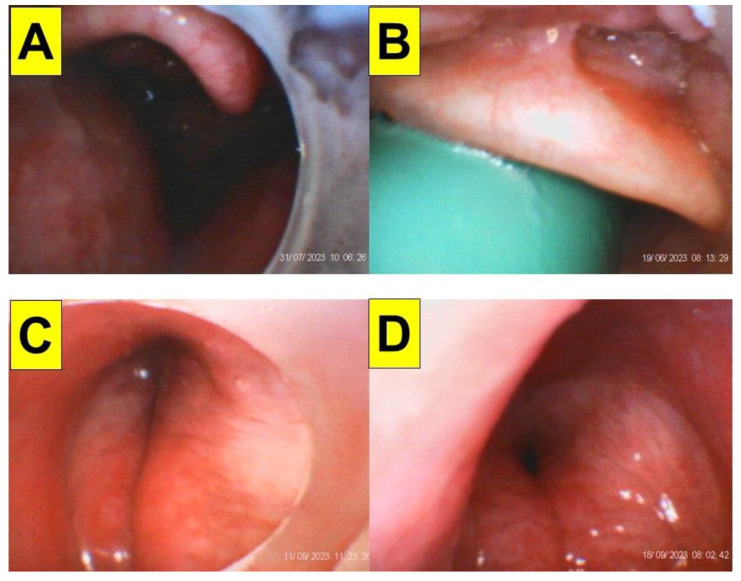
Examples of narrowing and collapse of airway in obese patients undergoing bariatric surgery. (**A**) BMI: 51.1 (Case 6, Table 1). Collapsed lateral pharyngeal wall with a concave epiglottis is prominent. (**B**) BMI: 48.0 (Case 10, Table 1) A suction-airway device was in advance placed under the anterior–posterior collapsed epiglottis. (**C**) BMI: 54.0 (Case 4, Table 1). Edematous and hypertrophic bilateral vestibular folds are observed. (**D**) BMI: 63.1 (Case 2, Table 1). Chronic hypertrophic laryngitis with bilateral arytenoid edema is observed. (See Appendix A).

**Figure 5 healthcare-12-01404-f005:**
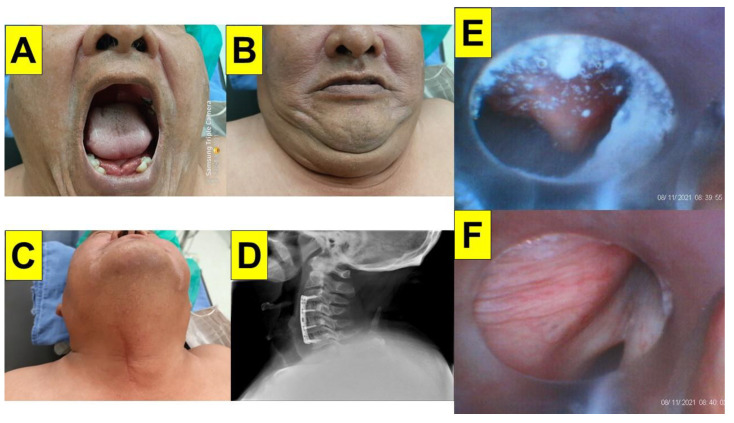
An obese patient with restricted cervical spine mobility undergoing bariatric surgery. BMI 45.4 (Case 12, Table 1). (**A**) Mouth-opening. (**B**) Compromised upper-lip bite test. (**C**) Stiff neck with limited extension. (**D**) Prior treatment with cervical spine fixation shown in X-ray. (**E**) An omega-shaped epiglottis obscured by saliva. (**F**) Glottis visualization. (See Appendix A).

**Figure 6 healthcare-12-01404-f006:**
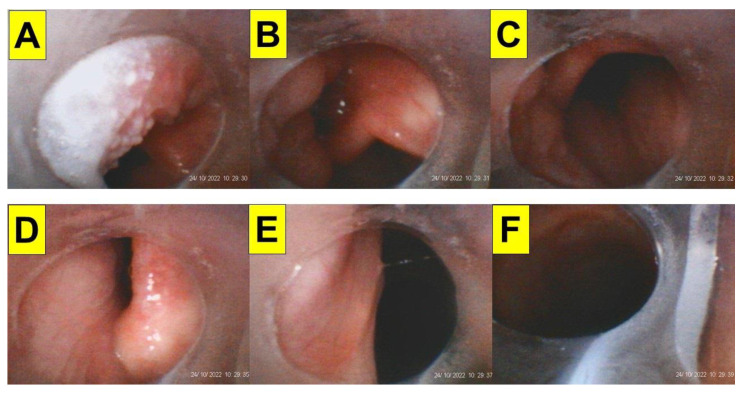
The performance of styletubation by a resident anesthesiologist. BMI 47.1 (Case 11, Table 1). (**A**–**C**) The collapsed and jammed soft tissues obstructed the upper airway. (**D**) Glottis. (**E**) Vocal cords. (**F**) Placement of endotracheal tube into trachea. (See Appendix A).

**Figure 7 healthcare-12-01404-f007:**
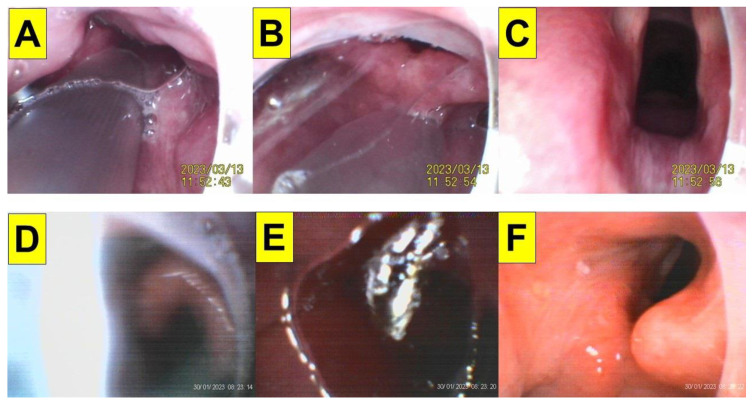
Disturbance of visualization by saliva/secretions during styletubation in obese patients. (**A**–**C**) BMI 36.6 kg/m^2^ (Case 18, Table 1). (**D**–**F**) BMI 63.6 kg/m^2^, (Case 3, Table 1). (See Appendix A).

**Figure 8 healthcare-12-01404-f008:**
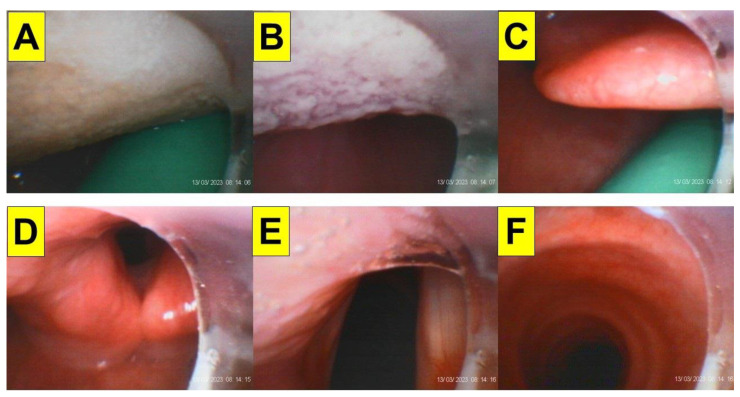
An easy solution to acquire clear airway visibility during styletubation in obese patients. BMI 37.8 kg/m^2^ (Case 17, Table 1). (**A**–**C**) A modified airway-suction tube unit (in green color) was used in advance to clear the saliva/secretions. (**D**–**F**) Clear visualization of glottis, vocal cords, and trachea (See Appendix A).

**Figure 9 healthcare-12-01404-f009:**
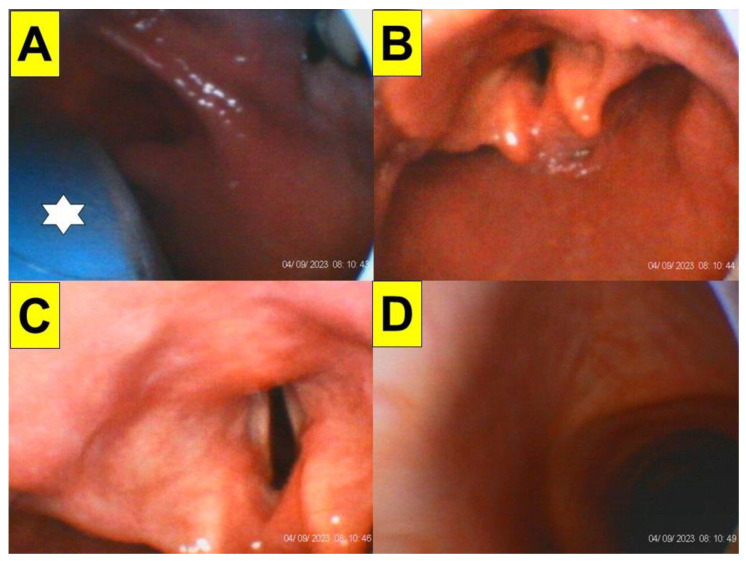
Styletubation assisted by laryngoscopy in obese patients. BMI 43.9 kg/m^2^ (Case-14, Table 1). (**A**) The white star denotes the VL blade in blue color. (**B**) An exposed glottis by laryngoscopy before styletubation is set out. (**C**,**D**) Views of vocal cords and trachea from styletubation (See Appendix A).

**Table 1 healthcare-12-01404-t001:** Styletubation applied on 20 obese patients undergoing bariatric surgery.

	Case 1	Case 2	Case 3	Case 4	Case 5	Case 6	Case 7	Case 8	Case 9	Case 10	Case 11	Case 12	Case 13	Case 14	Case 15	Case 16	Case 17	Case 18	Case 19	Case 20
Age/Gender (years old)	33/M	42/M	42/F	49/M	57/F	51/F	26/F	45/M	32/F	37/F	23/M	56/M	41/F	26/M	48/M	56/M	38/M	42/M	40/M	58/F
Height (cm)	158	165	161	180	153	160	160	171	157	162	176	159	158	166	175	170	178	166	174	166
Weight (kg)	258	172	165	175	120	131	130	147	119	126	146	115	111	121	124	112	120	101	109	96
BMI (kg/m^2^)	103	63.1	63.6	54	51.2	51.1	50.7	50.2	48.2	48	47.1	45.4	44.4	43.9	40.4	38.7	37.8	36.6	36	34.8
Surgery	LSG	OAGB	OAGB	LSG	LSG	OAGB	LSG	LSG	OAGB	OAGB	LSG	LSG	LSG	LSG	LSG	OAGB	LSG	LSG	LSG	LSG
Comorbidity	HTN/DM/OSAS	HTN/DM/OSAS	HTN/OSAS	HTN/OSAS	HTN/CAD/cirrhosis of liver	HTN/OSAS	Obesity/OSAS	CAD/CVA/OSAS	OSAS	HTN/DM	OSAS	HTN/DM/CVA	DM/asthma	OSAS	OSAS	HTN/DM/CAD	OSAS	OSAS/COPD	HTN/DM/OSAS	HTN/DM
ASA class	III	III	II	II	III	II	II	III	II	III	II	III	II	II	II	III	II	II	III	III
MMT	4	4	3	3	3	3	2	2	2	2	2	3	2			2	2	2	3	3
Interincisor distance	4.5	5.5	5	5.5	5	4.5	6	5	4	4	4.5	4.5	4	5	5.5	4.5	5	4.5	4	4.5
ULBT	2	2	1	2	2	1	1	1	2	1	1	2	1	1	2	1	1	1	2	1
Thyromental distance (cm)	8	6	7	9	8	7	9	9	9	9	8	6	7	7	9	9	7	7	6	8
Sternomental distance (cm)	17	15	16	15	14	16	20	18	16	17	16	15	16	18	13	16	17	16	16	18
Neck circumference (cm)	54	54	42	50	37	47	39	48	45	39	50	49	45	47	44	46	44	41	44	38
Neck fat deposition/soft tissue loading on the upper airway	moderate	moderate	mild	moderate	moderate	moderate	Mild	mild	mild	Mild	mild	moderate	mild	mild	mild	mild	mild	mild	mild	mild
Waist circumference (cm)	200	166	155	150	142	155	132	143	142	133	143	133	114	124	140	115	124	112	119	110
Hip circumference (cm)	202	154	158	148	144	150	146	126	156	134	148	126	140	128	124	115	118	120	120	126
LQS grading	2	2	2	2	2	1	2	1	1	2	2	2	2	1	2	2	2	2	2	1
POGO (%)	100	100	100	100	100	100	100	100	100	100	100	100	100	100	100	100	100	100	100	100
Time to intubation	12 s	14 s	18 s	22 s	6 s	10 s	8 s	11 s	5 s	5 s	11 s	13 s	9 s	6 s	18 s	8 s	12 s	22s	20 s	5 s
Expected cause of prolonged time of tracheal intubation											Performed by a third- year resident	Prior cervical spine surgery, copious secretions						For demonstration purpose	For demonstration purpose	
First-pass success	yes	yes	yes	yes	yes	yes	yes	yes	yes	yes	yes	yes	yes	yes	yes	yes	yes	yes	yes	yes
Subjective easiness	easy	easy	easy	easy	easy	easy	easy	easy	easy	easy	easy	easy	easy	easy	easy	easy	easy	easy	easy	easy
Airway-related complications	none	none	none	none	none	none	none	none	none	none	none	none	none	none	none	none	none	none	none	none
Supplemental video-clips	Video S1	Video S2	Video S3	Video S4	Video S5	Video S6	Video S7	Video S8	Video S9	Video S10	Video S11	Video S12	NA	Video S13	Video S14	Video S15	Video S16	Video S17	Video S18	Video S19

BMI: body mass index; LSG: laparoscopic sleeve gastrectomy; OAGB: one-anastomosis gastric bypass; HTN: hypertension; DM: diabetes mellitus; OSAS: obstructive sleep apnea syndrome; CAD: coronary artery disease; CVA: cerebrovascular accident; ASA: American Society of Anesthesiologists; MMT: modified Mallampati test; ULBT: upper lip bite test; LQS grading: Luk–Qu–Shikani grading; POGO scale: percentage of glottic opening scale.

## Data Availability

The original contributions presented in the study are included in the article/Appendix A; further inquiries can be directed to the corresponding author.

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
