# Peer review of "Structured Routine Use of Styletubation for Oro-Tracheal Intubation in Obese Patients Undergoing Bariatric Surgeries—A Case Series Report"

_healthcare, 2024, doi:10.3390/healthcare12141404_

Round 1

Reviewer 1 Report

Comments and Suggestions for Authors

This article reports on a case series of obese patients who underwent bariatric surgery, examining practitioner’s experiences with upper airway management using stylet-guided oro-tracheal intubation. The authors state that the results of employing this procedure were satisfactory and propose that the styletubation technique can be a preferred method for tracheal intubation in obese patients undergoing bariatric surgery.

The topic is interesting due to the scarcity of studies exploring alternative intubation methods to direct laryngoscopy, especially in cases of difficult upper airways, particularly among obese patients undergoing bariatric surgery. Moreover, the authors clearly explained the procedures with numerous illustrative figures and delineated the advantages of styletubation as well as its limitations, which are almost comparable to those of videolaryngoscopy. I find that this manuscript may be suitable for publication after a minor revision. Below are some comments:

Line 34-43: please, add reference(s)

Line 42: please, explain briefly, why BMI may not be comparative to the same degree of fat distribution in different individuals and specify if other method(s) are available for this purpose.

Table 2 needs further clarification: specify whether BMI values represent a mean or a median; provide the complete nomenclature of POGO.

Author Response

Response to the Reviewer-1

This article reports on a case series of obese patients who underwent bariatric surgery, examining practitioner’s experiences with upper airway management using stylet-guided oro-tracheal intubation. The authors state that the results of employing this procedure were satisfactory and propose that the styletubation technique can be a preferred method for tracheal intubation in obese patients undergoing bariatric surgery. The topic is interesting due to the scarcity of studies exploring alternative intubation methods to direct laryngoscopy, especially in cases of difficult upper airways, particularly among obese patients undergoing bariatric surgery. Moreover, the authors clearly explained the procedures with numerous illustrative figures and delineated the advantages of styletubation as well as its limitations, which are almost comparable to those of videolaryngoscopy. I find that this manuscript may be suitable for publication after a minor revision. Below are some comments:

Comment-1: Line 34-43: please, add reference(s)

Response-1: The information are from WHO references and others. However,  in compliance to the other reviewer’s requirements, we have to reduce the contents of the original version of the paper and cope with the format of a “case-series report”. Therefore, the background introduction of obesity and those with BMI criteria are now deleted. Thanks for your suggestion/.

Comment-2: Line 42: please, explain briefly, why BMI may not be comparative to the same degree of fat distribution in different individuals and specify if other method(s) are available for this purpose.

Response-2: Same as the response-1, we have to delete this paragraph related to the definition of obesity and the role of BMI. Regarding the previous statement of “….it should simply be regarded as a rough guide because BMI may not be comparative to the same degree of fat distribution in different individuals.”, what we like to emphasize was that even with same BMI in the obese subjects, the impact of fat tissues distribution could be different for “airway management”. What one should be alert is the proportion and degree of fat distribution around the neck region (e.g., not the hip or waist area) where the upper airway structure could be serious affected for airway management. Thanks for your point.   

Comment-3: Table 2 needs further clarification: specify whether BMI values represent a mean or a median; provide the complete nomenclature of POGO.

Response-3: Due to other reviewer’s comments, we unfortunately have to delete the Table-1, Table-2 and Table-3 in the revised version of the manuscript. The value of the BMI were the mean value. The nomenclature of POGO has been spelled out in the footnote of the new Table 1. “POGO scale: percentage of glottic opening scale”. Thanks for your comments.

The authors appreciate your excellent points and constructive comments and suggestions.

Reviewer 2 Report

Comments and Suggestions for Authors

The paper addresses a pertinent topic as bariatric surgery has recently expanded. However, the manuscript's primary issue is its non-conformity to any recognized publication type. It simultaneously attempts to be a brief case series report of 20 patients and a narrative review, resulting in a disjointed structure filled with fragmented information and extensive tables. In my view, the author could have opted for a brief research article format, utilizing the 20 well-documented cases: an Introduction to succinctly incorporate parts of the narrative review, Materials and Methods to clarify the study's design, informed consent if applicable, approval details, and a detailed presentation of the tool (technical characteristics, manufacturer), along with a step-by-step description of the technique, including whether the approach was midline or retromolar. The Results section should list the results and any complications, and the Discussion could compare the advocated method's benefits to videolaryngoscopy and other previously discussed topics. Tables 1, 2, and 3 should be removed, and Table 4 should be simplified to show mean/median values instead of individual patient data. Given that the majority of tracheal intubations at your institution are performed with an intubating stylet, this case series could precede a larger randomized trial. I suggest the paper be revised to align with its title: a structured case series report without attempts at a narrative review. All general information should be confined to the Introduction, with controversies, future directions, and the potential advantages of stylet intubation incorporated into the Discussion. The text needs to be more concise and precisely structured, avoiding any resemblance to a narrative review.

Author Response

Response to the Reviewer-2

Comment-1: The paper addresses a pertinent topic as bariatric surgery has recently expanded. However, the manuscript's primary issue is its non-conformity to any recognized publication type. It simultaneously attempts to be a brief case series report of 20 patients and a narrative review, resulting in a disjointed structure filled with fragmented information and extensive tables. In my view, the author could have opted for a brief research article format, utilizing the 20 well-documented cases: an Introduction to succinctly incorporate parts of the narrative review, Materials and Methods to clarify the study's design, informed consent if applicable, approval details, and a detailed presentation of the tool (technical characteristics, manufacturer), along with a step-by-step description of the technique, including whether the approach was midline or retromolar. The Results section should list the results and any complications, and the Discussion could compare the advocated method's benefits to videolaryngoscopy and other previously discussed topics.

Response-1: The authors appreciate the reviewer’s excellent and very constructive comments! We revise the format and content of our manuscript accordingly.

  • We absolutely agree with the comment regarding the mixed type of case report and narrative review in our previous version of the manuscript. Therefore, we have removed the unnecessary review contents and focused on the case report purpose!
  • We revised the manuscript in a traditional format as: “Introduction”, “Case Presentation”, and “Discussion”.
  • In Introduction section, there are only 4 paragraphs now. (info regarding obesity, bariatric surgery, roles of DL/VL, and potential role of styletubation for tracheal intubation in bariatric surgery.
  • In Case Presentation, we added the info regarding the technical part (instrument, operational info). We include the info about the written permission from our IRB for the ethics requirement (Institutional Review Board Statement:This study was conducted according to the guidelines of the Declaration of Helsinki and was approved by REC, Hualien Tzuchi Hospital (approved letter number: CR113-08)). Technical details have been described including the “midline approach”. Reference 32-35 are also provided for detailed tech info.
  • In Discussion section, we described the adv/disadv and pearls/pitfalls of styletubation in comparison to DL/VL in the scenarios of obesity and bariatric surgery.
  • The length of the current version of case report has now downsized and the references number has also been reduced.

Comment-2: Tables 1, 2, and 3 should be removed, and Table 4 should be simplified to show mean/median values instead of individual patient data. Given that the majority of tracheal intubations at your institution are performed with an intubating stylet, this case series could precede a larger randomized trial. I suggest the paper be revised to align with its title: a structured case series report without attempts at a narrative review. All general information should be confined to the Introduction, with controversies, future directions, and the potential advantages of stylet intubation incorporated into the Discussion. The text needs to be more concise and precisely structured, avoiding any resemblance to a narrative review.

Response-2:

  • Table-1, Table-2, and Table-3 have now been deleted.
  • Table-4: Now renew as Table-1. Regarding the info in the Table-1, we wish to keep the raw data from each patient. The reason is that we provide the imaging info at the same time. Perhaps the readers can find more useful to link the images to the individual patient’s profile (e.g., the degree of DA related to his/her predicted airway profile). In that sense, the mean/median probably can not clearly reflect the individual airway scenario.
  • In compliance with your suggestion, we have already removed all the intentions of “narrative review” contents in the revised version.
  • The word of “structured” has been added into the new title of the paper. As you indicated, we plan to continue this story with a retrospective study in the near future, with much more cases number.
  • Again, the authors really appreciate your time and efforts to provide excellent and constructive suggestions and comments on our manuscript.

Round 2

Reviewer 2 Report

Comments and Suggestions for Authors

I deem the manuscript fit for publication following revision and I congratulate the authors.